# Multi-Species Host Use by the Parasitoid Fly *Ormia lineifrons*

**DOI:** 10.3390/insects14090744

**Published:** 2023-09-05

**Authors:** Kyler J. Rogers, Oliver M. Beckers

**Affiliations:** Department of Biological Sciences, Murray State University, 1101 Biology Building, Murray, KY 42071, USA; krogers31@murraystate.edu

**Keywords:** *Neoconocephalus*, Tachinidae, katydid, multivoltine, arms race, parasitism

## Abstract

**Simple Summary:**

Species interact with each other in many ways, such as predation, mutualism, or competition, ultimately affecting each other’s evolution, reducing the negative effects of these interactions and/or increasing the positive ones. We studied a parasitic fly that uses different katydid species as hosts for its larvae sequentially at different times during its breeding season. The fly larvae kill the host when they emerge, setting up multiple arms races between the fly and each of the hosts. The hosts are selected to ameliorate the negative impact of parasitism, whereas the parasitoid is selected to use all host species well for the population to persist. We compared host use and the ability of fly pupae to develop into adults across four katydid hosts. The parasitism rate varied between ~14% and 73%, yet host use was similar for many larval measurements across hosts. However, one species was a particularly poor host for the fly based on fly pupa development. We suggest that currently, this poor host strongly influences the evolution of the fly in this arms race to improve its utilization of this host. Understanding how species interact and affect each other is important to predict their evolution and manage species of human relevance, such as parasites, plant pests, or vectors of disease.

**Abstract:**

Antagonistic species relationships such as parasitoid/host interactions lead to evolutionary arms races between species. Many parasitoids use more than one host species, requiring the parasitoid to adapt to multiple hosts, sometimes being the leader or the follower in the evolutionary back-and-forth between species. Thus, multi-species interactions are dynamic and show temporary evolutionary outcomes at a given point in time. We investigated the interactions of the multivoltine parasitoid fly *Ormia lineifrons* that uses different katydid hosts for each of its fly generations sequentially over time. We hypothesized that this fly is adapted to utilizing all hosts equally well for the population to persist. We quantified and compared the fly’s development in each of the four *Neoconocephalus* hosts. Cumulative parasitism rates ranged between ~14% and 73%, but parasitoid load and development time did not differ across host species. Yet, pupal size was lowest for flies using *N. velox* as a host compared to *N. triops* and other host species. Successful development from pupa to adult fly differed across host species, with flies emerging from *N. triops* displaying a significantly lower development success rate than those emerging from *N. velox* and the other two hosts. Interestingly, *N. triops* and *N. velox* did not differ in size and were smaller than *N. robustus* and *N. nebrascensis* hosts. Thus, *O. lineifrons* utilized all hosts but displayed especially low ability to develop in *N. triops*, potentially due to differences in the nutritional status of the host. In the multi-species interactions between the fly and its hosts, the poor use of *N. triops* may currently affect the fly’s evolution the most. Similarities and differences across host utilization and their evolutionary background are discussed.

## 1. Introduction

Species interactions, especially antagonistic ones such as competition, predation, and parasitism, introduce strong selective pressures on the interacting species [1,2]. As a result, both species evolve traits to mitigate the negative effects resulting from these kinds of interactions. Such coevolutionary arms races have led to a broad range of adaptations in behavior, morphology, and physiology in a range of species, e.g., [3,4,5,6]. In this evolutionary back-and-forth between species, at a given point in time, one species could adaptively speaking be the leader, while the other one is the follower, fueling further rounds of the arms race. The evolutionary outcomes become even more complex if more species interact sequentially, with one species (e.g., predator) being required to overcome the counter-adaptations of two or more species at the same time. These multi-species arms races provide multiple opportunities to detect evolutionary leaders and followers and provide important insights in the evolutionary dynamics of arms races.

Parasitoids typically use more than one host species [7]. Two of the main determinants of how many species a parasitoid can use, i.e., its host range, are host taxonomy and ecology [8,9]. Parasitoids may attack taxonomically closely related hosts because they share similar physiological and mechanical defense mechanisms that the parasitoid can overcome with the same set of evolved adaptations [7]. As a result, endoparasitic koinobionts, i.e., parasitoids that develop inside hosts that keep growing, moving, and feeding, have narrower host ranges and are more specialized because of the more intimate physiological and biochemical connection to their hosts compared to ectoparasitoid idiobionts that paralyze their hosts or use sessile host stages (e.g., eggs, pupae) and do not deal as intimately with host defenses [7,8]. The host’s ecology can affect the parasitoid’s host range in many ways as well. For example, host phenology needs to overlap with that of the parasitoid for the two species to interact. Also, if the host uses plant chemicals in defense against parasitoids, this may affect which species are utilizable by the parasitoid [7]. Host species that share the same habitat or food plant species frequently also share the same parasitoids that are attracted by odors related to the habitat or plants, e.g., [10].

*Ormia* (Diptera: Tachinidae) is a diverse genus of parasitoids that use mating signals of primarily orthopteran hosts, such as crickets and katydids, to identify and find their hosts [11]. These koinobiont endoparasitoids place their first instar larvae on or near a host [12], which then enter the host, develop inside the host, and emerge from the host within 10 days or less [12,13]. The larvae pupate after emergence and develop into adult flies after about two weeks [14,15]. The larvae kill the host when they emerge, substantially reducing the life span and signaling time of the hosts [12,13]. Thus, selective pressure on evolving counter-adaptations is expected to be high for both species in this arms race, and the relatively short generation times of these insects should further facilitate rapid evolutionary responses [16].

A large body of work has provided exceptional insights into the parasitoid/host arms race of *Ormia ochracea* Bigot, 1889 (Diptera: Tachinidae), which uses field crickets as hosts, e.g., [13,17,18]. In these interactions, each fly population typically uses a single host species in their geographical range [19], and the arms race takes place between one fly and one host. In contrast, very little is known about *Ormia lineifrons* Sabrosky, 1953 (Diptera: Tachinidae), which uses katydids as hosts [13]. We recently showed that *O. lineifrons* is multivoltine in Kentucky, and each generation uses different *Neoconocephalus* species [15] that are univoltine and call at different times of the year as hosts. More specifically, the first fly generation uses *N. triops* Linnaeus, 1758 (Orthoptera: Tettigoniidae), their offspring or the second generation uses *N. velox* Rehn & Hebard, 1914 (Orthoptera: Tettigoniidae), and the third generation uses partially overlapping *N. nebrascensis* Bruner, 1891 (Orthoptera: Tettigoniidae) and *N. robustus* Scudder, 1863 (Orthoptera: Tettigoniidae) katydids as host species [15]. Thus, each katydid species serves as a steppingstone, allowing the multivoltine fly to reproduce throughout the year, similar to the multivoltine parasitoid wasps, *Utetes anastrephae* Viereck, 1913 and *Doryctobracon areolatus* Szépligeti, 1911 (both Hymenoptera: Braconidae), that use different generations of the pest fruit fly *Anastrepha obliqua* Macquart, 1835 (Diptera: Tephritidae) throughout the year [20]. *O. lineifrons* uses closely related host species that likely share similar physiological and mechanical defense mechanisms. Additionally, the hosts’ phenology is of importance for *O. lineifrons*, since each univoltine host coincides with a different generation of the fly [15]. Thus, these sequential species interactions between *O. lineifrons* and its *Neoconocephalus* hosts are different from those of one parasitoid using multiple hosts at the same time. In the first case, poor utilization of one host species can negatively affect subsequent fly generations (e.g., population size, genetic diversity; sensu [20]), whereas in the second case, the parasitoid could compensate for negative effects by exploiting other simultaneously available species. Thus, in this multi-species interaction, *O. lineifrons* is expected to be selected to compensate for the counter-adaptations of each host species, because the success of utilizing a given host species affects not only the current but also subsequent generations of the parasitoid.

In this study, we compared host use of *O. lineifrons* fly larvae across the four *Neoconocephalus* host species (*N. triops, N. velox, N. nebrascensis, N. robustus*) in Kentucky (U.S.) that we identified previously [15]. Since the fly uses closely related host species that are likely similar in their defense mechanisms and selection is expected to be high in order to utilize all hosts for the fly population to persist, we hypothesized that *O. lineifrons* is adapted to utilizing all four host species equally well. Thus, we predicted that host use and the ability to develop would not differ across host species. To test our hypothesis, we collected parasitized males of all four host species and compared multiple measures of larval development across hosts. This parasitoid/host system offers a valuable opportunity to better understand the adaptive capabilities and limitations of a parasitoid exploiting multiple host species in a complex arms race and learn about the evolutionary dynamics of arms races.

## 2. Materials and Methods

### 2.1. Host Collection and Animal Care

We collected calling katydids of each species within a 30-mile radius around Murray, KY between March and October of 2019 (n = 199) and 2020 (n = 187). We only collected male katydids since female katydids do not call, and we have not observed parasitism of females in the past (OMB personal observation). Each collected katydid was placed in a cage (Lee’s Aquarium and Pet Products, San Marcos, CA, USA; 15.57 cm × 23.19 cm × 15.25 cm, height × length × width) with its lid lined with an insect screen (Small bug screen, #14151, M-D Building Products, Oklahoma City, OK, USA) on the underside to prevent roaming fly larvae from escaping the cage. Emerged larvae typically pupate within one or a few hours (OMB personal observations). We supplied the katydids with ad libitum organic apple slices, cricket food (Fluker’s High-Calcium Cricket Diet, Fluker Farms, Port Allen, LA, USA) and rolled oats for food, and water gel (Tasty Worms Nutrition, Inc., Longwood, FL, USA) as the main source of water. Additionally, we sprayed the cages daily with water. We kept the katydids in an incubator (PR505755L; Thermo Fisher Scientific, Waltham, MA, USA) with a light/dark cycle of 15.5/8.5 h and temperatures of 26/22 °C (day/night), respectively. The incubator had a relative humidity between 65% and 85%. The day length and temperatures roughly corresponded to a long summer day in Kentucky.

### 2.2. Parasitism Measurements

*Ormia lineifrons* larvae typically emerge from the katydid within nine days [13]. We checked each cage daily for the presence of fly larvae or pupae and katydid mortality. If katydids were found dead, we checked the food and water gel containers for larvae or pupae. All dead katydids were kept in the incubator for 24 h to allow time for any remaining larvae inside the host to emerge. We then dissected the thorax and abdomen of the katydid under a dissecting microscope (Stemi 1000, Zeiss, Oberkochen, Germany) and checked for larvae inside the host. Any larvae found inside dead hosts were deceased, and we interpreted these larvae as evidence for superparasitism, i.e., multiple larval deposition events on the same host at different times [21,22]. We determined the cumulative parasitism and superparasitism rate of all four host species over the two years of collection. We previously described the weekly parasitism rates in this population [15]. The flies typically attack each host for a few weeks, and maximum parasitism rates reach 100% in *N. triops* and *N. velox* and between 50 and 60% in *N. robustus* and *N. nebrascensis* [15]. To determine the parasitoid load, we counted all pupae that emerged and larvae that were found inside the host. We kept unparasitized katydids in the laboratory for at least six weeks following collection before we froze them (−20 °C for 48 h).

### 2.3. Fly Care and Measurements

Each pupa found in a cage was weighed (Ohaus; Model PA84, Parsippany, NJ, USA) and then transferred to a 50 mL centrifuge tube on the same day as emergence and placed at the base of the tube (Falcon, Corning, Tamaulipas, Mexico) on cotton (#3166, Dynarex Corporation, Orangeburg, NY, USA) that was sprayed with a saturated 0.4% methylparaben solution to restrict fungal growth and provide moisture [15]. The caps of the tubes had holes allowing for gas exchange. We kept the centrifuge tubes upright in the same incubator in the same conditions as the collected katydids (see above). We checked daily for fungus growth and cotton moisture, and sprayed Methylparaben as needed. We noted fly emergence from the pupae and calculated the development time it took the larva to emerge from its host until the day it developed into an adult fly. We noted for each host how many of the larvae matured into adult flies as a measure of pupal development success rate. It takes about two weeks for flies to develop from pupae to adulthood [15,21], and we kept pupae for at least four weeks before we categorized pupal development as a failure.

### 2.4. Host Measurements

We measured host size as an indicator for the potential amount of available food for the larvae. More specifically, we measured the hind right femur length of katydid males (only in 2020) within ±0.01 mm using digital calipers (Digimatic, Mitutoyo, Kawasaki, Japan) as a proxy for host size [23].

### 2.5. Data Analysis

We used analysis of variance (ANOVA) models to compare the host sizes and parasite loads across species. For the analysis of development time, we used an ANOVA with ‘species’, ‘parasitoid load’ and the interaction of the two as fixed effects. To compare development time between males and females among species, we ran an ANOVA with ‘sex’, ‘species’, and their interaction as fixed effects. For katydids that were hosts to more than one developing pupae, we averaged the developmental times across the clutch of successfully developing pupae into adult flies, resulting in one data point per host [15]. For the development success rate of pupae, i.e., the proportion of pupae developing to adulthood per clutch, we used a logistic regression model with ‘species’ as the main effect. Note that for this success rate, we calculated for each host individual the proportion of fly pupae that successfully developed into adult flies relative to all the larvae that the host harbored. We compared the mass of pupae that successfully developed into adults using a linear model with ‘species’, ‘parasitoid load’, and the interaction as fixed effects. Across all analyses, we removed non-significant interactions from our models and used post hoc Tukey HSD tests to compare significant main effects. None of the data violated normality and homogeneity of variance assumptions of parametric statistical tests except for the data related to the comparison of developmental time between male and female flies. These data violated the normality assumption of parametric models, which could not be augmented by data transformations. We used bootstrapping (‘lmboot’ package in R; 1000 bootstrap samples) rather than the F-distribution to determine the *p*-value of the reported ANOVA, due to the deviations from normality of the residuals, as indicated in the normal probability plot. Significant models were significant at *p* < 0.05. We used the software R (Version 1.4.1106) and JMP (version 16.2.0 for MAC) to run the models.

## 3. Results

### 3.1. Parasitism Rates and Superparasitism

The cumulative parasitism rate for *N. triops* males across both years was 31.8% (parasitized males/all collected male *N. triops*; n= 49/154; see also Table 1), and of the 49 parasitized males, 8% were superparasitized (n = 4). The cumulative parasitism of *N. velox* males was 72.7% (n = 24/33), and of the 24 parasitized males, 25% were superparasitized (n = 6). *Neoconocephalus robustus* and *N. nebrascensis* had cumulative parasitism rates of 14.3% (n = 12/84) and 17.5% (n = 7/40), respectively. We did not detect superparasitism in these two host species.

### 3.2. Pupal Development

The parasitoid load did not differ among host species (ANOVA, F_3,83_ = 2.04, *p* = 0.11; Table 1). Similarly, pupal development time did not differ across the four host species (ANOVA, F_3,54_ = 0.400, *p* = 0.756), and was not affected by parasitoid load either (ANOVA, F_1,54_ = 0.001, *p* = 0.971; Table 1). However, across species, female pupae took longer to develop into adult flies compared to male offspring (ANOVA, F = 4.27, *p* = 0.046; Figure 1).

Pupal mass of insects that successfully developed into adult flies differed significantly among species (ANOVA, F_3,39_ = 10.70, *p* < 0.001), with the mass of pupae that emerged from *N. velox* being significantly lighter than those of the other three host species (all post hoc Tukey HSD tests: *p* ≤ 0.003; Figure 2). Pupal mass did not differ among *N. triops*, *N. robustus*, and *N. nebrascensis* (all post hoc Tukey HSD tests: *p* ≥ 0.83). Across species, pupal mass decreased significantly with parasitoid load (ANOVA, F_1,39_ = 33.98, *p* < 0.001; Figure 3). We detected the same significant patterns when analyzing the mass of all pupae, i.e., those that did and did not develop into adult flies (see Appendix A).

Pupal developmental success to adulthood differed significantly across species (logistic regression, df = 3, X^2^ = 12.32, *p* = 0.0064), with larvae emerging from *N. triops* displaying a significantly lower success rate than those emerging from *N. velox* (post hoc Tukey HSD test: z = 2.85, *p* = 0.02, Figure 4). All other comparisons across species did not indicate any significant differences (all post hoc Tukey HSD tests: *p* ≥ 0.099).

### 3.3. Host Size

Host species differed significantly in size (ANOVA, F_3_,_105_ = 94.60, *p* = 0.0001; Figure 5). Specifically, *N. robustus* was larger than all other species (post hoc Tukey HSD tests: *p* = 0.0001 for each comparison) and *N. nebrascensis* was larger than *N. velox* and *N. triops* (post hoc Tukey HSD test: *p* = 0.0001 for *N. triops*, *p* = 0.002 for comparison with *N. velox*). There was no difference in the mean size of *N. velox* and *N. triops* males (post hoc Tukey HSD test: *p* = 0.76).

## 4. Discussion

The larvae of the parasitoid *O. lineifrons* successfully utilized all four host species. Parasitism rates varied by host, covering a range from 14% to 73%. Across hosts, parasite load and developmental time did not differ; however, fly pupae emerging from *N. velox* were significantly smaller than those emerging from the other three hosts. Yet, this lower mass did not negatively affect pupal development rate to adult flies, which was similar or even better (*N. triops*) compared to other host species. Below, we discuss the possible implications of the developmental differences and similarities of the fly across host species.

### 4.1. Differences in Fly Development and Possible Explanations

Pupal size differed significantly across species, with the smallest pupae emerging from *N. velox* hosts. Typically, the juvenile size of parasitoids is negatively correlated to competition inside the host and positively correlated with host size [24]. We found that pupae that emerged from hosts that were parasitized by multiple larvae were smaller (see Figure 3). However, the relationship between the parasitoid and host size was more complex. *Neoconocephalus velox* had a similar size as *N. triops*, and both species were smaller than *N. robustus* and *N. nebrascensis*. Even though the smaller host size can explain the lower pupal mass of *N. velox* in comparison to *N. nebrascensis* and *N. robustus*, it cannot explain why the pupae were smaller compared to the similar-sized *N. triops*. The pupal developmental success rate to adulthood differed across species, with those emerging from *N. velox* displaying a higher success rate than those of *N. triops* (i.e., 50% vs. 70%). Note that we found this lower rate of successful pupal development in *N. triops* not only in the Kentucky but also in the Florida population (50%; [21]), supporting the possibility that pupal development of *O. lineifrons* is indeed less successful when using *N. triops* as the host. This notable difference in the fly’s developmental success rate related to *N. velox* and *N. triops* hosts warrants further discussion. Host size is a crucial factor for parasitoid fitness ([7,25], but [26]), since parasitoids can derive more nutrition from bigger hosts (sensu, [27,28,29]). However, host size did not differ between *N. triops* and *N. velox*, possibly indicating that it is not the amount of food that the larvae can acquire that affects their growth, but possibly that *N. triops* is a lower-quality host than *N. velox.* For example, hosts may differ in the quality of the provided nutrients or nutrient compositions (sensu [26]) that *O. lineifrons* larvae receive from feeding on them. The closely related parasitoid fly *Ormia ochracea* uses field crickets as hosts [19,30], and the larvae feed primarily on the host’s fat body, thoracic muscles, and to a lesser extent the reproductive organs [12]. It is possible that *N. velox* provides a higher concentration of the primary nutrients stored in these organs (i.e., fat, protein) than *N. triops*, better preparing the developing larvae for metamorphosis. Differences in life history between *N. triops* and *N. velox* may also explain a potential difference In their nutritive values. *Neoconocephalus velox* overwinters in the egg stage, and hatches and matures throughout the spring and summer, developing during times when plenty of food (e.g., grass and grass seeds) is available. In contrast, *N. triops* overwinters as an adult before finishing maturation and starting to reproduce in the following spring [31]. *Neoconocephalus triops* accumulates fat reserves in preparation for the several-month-long diapause [31]. However, diapausing as an adult may deplete the fat reserves in *N. triops,* and maturation of the reproductive organs and calling takes place when few resources are available (mid-March in Kentucky; OMB’s personal observation), which may prevent these hosts from accumulating much fat tissue. Thus, differences in fat content and possibly other nutritive components between *N. triops* and *N. velox* may explain the different developmental success rate of the pupae emerging from these two hosts.

It is also possible that *N. triops* has evolved more effective counter-adaptations to fly development than *N. velox*, exerting more stress on the developing larvae that pupate. For example, stronger immune responses while the larvae develop inside the host may weaken the larvae and reduce their ability to finish pupal development after they emerge (sensu [32,33]). Note that we did not test the survival of *O. lineifrons* larvae inside the hosts, which would have required knowledge of parasitoid load when they were collected in the field and killing the katydids before the pupae emerged from the host. In future experiments, larval developmental success will be measured using artificial host infections, which may indicate additional differences in utilization of the different hosts.

### 4.2. Similarities in Host Use

The parasitoid load did not differ significantly across host species, even though host quality seemed to differ, as discussed in the previous section. Female parasites typically adjust the clutch size deposited on hosts based on the host’s size or quality to achieve maximum fitness [34]. However, to gain information of host quality, the parasite needs to closely interact with its host. Tachinid flies such as *Ormia* lack a rigid ovipositor [35] and typically place their mobile planidia larvae close to their host or eject them onto their hosts [12,30] without close interaction with the host, possibly explaining why *O. lineifrons* did not adjust its clutch size (sensu [12]) and parasitized already infested hosts, as evidenced by superparasitism. Besides this proximate reason, it is also possible that this rather consistent parasite load of one to two larvae per host may maximize the fitness returns across hosts and represents an evolved response.

Juvenile development of parasitoids frequently differs related to host size and quality (review in [7]). However, we did not detect any difference in pupal development time across hosts, even though the hosts differed in size and likely in quality. *Ormia lineifrons* exploits different transient host species at different times of the year and is highly synchronized with the time of their occurrence [15], possibly selecting against much variation in development time to better synchronize their phenology with that of each host. We previously reported that flies typically parasitize their host species after they have been reproductively active [15], and therefore longer pupal development times could reduce the chances of encountering hosts due to the host’s aging or possible predation events. We detected slightly longer development times of female flies compared to male flies across species, which is in line with other studies showing that females take longer to develop than males, due to their larger size [28,36,37].

### 4.3. Co-Evolution

Our study provides insights into a highly complex evolutionary arms race between one parasitoid and its multiple host species. Since *O. lineifrons* uses sequentially different hosts at different times of the year due to its multivoltine life history, each fly generation directly depends on each of the exploited host species to finish its own reproduction. This intimate relationship with all of its hosts requires the fly to compensate for any of the host’s counter-adaptations for the fly population to remain stable [15,20].

The prevalence of parasitism varied among host species, with *N. velox* and *N. triops* displaying the highest prevalence (72.7% and 31.8%, respectively) and *N. robustus* and *N. nebrascensis* the lowest prevalence (14.3% and 17.5%, respectively), suggesting that the fly exerts different selective pressures on the hosts. Because of its high cumulative parasitism, *N. velox* is expected to be in an especially strong evolutionary arms race with the fly, selecting for efficient counter-adaptations against parasitism. It is possible that the smaller mass of pupae emerging from *N. velox* represents such a counter-adaptation of the host. However, the smaller pupal mass did not lower the rate of pupal development into adult flies, possibly indicating previous rounds of adaptations and successful counter-adaptations between *N. velox* and *O. lineifrons*, a hypothesis that requires further testing.

Our study suggests that *N. triops* is seemingly the least utilizable host for *O. lineifrons* in terms of pupal development, possibly affecting the fly’s evolution currently more than the other host species. Despite the poor pupal development in *N. triops*, no other katydid species is available for the fly as host in early spring (see below), requiring the fly to utilize this host. As the first host to be utilized by the fly in the year, *N. triops* affects subsequent fly generations in terms of population size and genetic variability (sensu [20]), and likely plays an elevated role in the evolution of the fly, which will be tested in the future.

The species interaction of *O. lineifrons* and *Neoconocephalus* hosts is different from a parasite using multiple hosts at the same time (e.g., parasites of primates: [38], parasites of domestic animals: [39], parasites of fish: [40]) or parasite populations using different hosts in each of their geographic ranges, such as *O. ochracea*, e.g., [19]. In our parasitoid/host relationship, each host can evolve its own counter-adaptations, but any of these adaptations may also affect the evolution of other hosts. For example, one host calling from more concealed positions (e.g., *N. bivocatus*; [41]) may select for more efficient or persistent host searching behavior in the fly, facilitating also more efficient detection of other hosts as a by-product. Thus, the evolution of species within one arms race can affect the evolution of species in another arms race, which becomes even more complex in *O. lineifrons* and its four *Neoconocephalus* hosts.

Both *O. lineifrons* [42] and *N. triops* [43,44] are tropical species that extended their range into temperate North America. In addition, both species share similar phenologies with adult activity in the early spring. These features put *O. lineifrons* and *N. triops* in an especially intimate historical relationship with each other, and it is possible that *N. triops* could have been the host that allowed *O. lineifrons* to expand its range from the tropics into temperate climates (sensu [45]). This in turn may also indicate that *N. triops* has been in an evolutionary arms race with *O. lineifrons* for much longer than with the temperate *N. velox* [42] and other temperate host species (i.e., *N. nebrascensis*, *N. robustus*), possibly having provided this katydid more time to evolve better counter-adaptations against *O. lineifrons*.

No matter the reason, in contrast to the expectation that *O. lineifrons* would be able to exploit all hosts that are closely related and likely share similar host defenses, the fly showed poorer pupal development rates in one host (*N. triops*) compared to the others. This result indicates that *N. triops* in Kentucky currently seems to be one step ahead in the arms race with *O. lineifrons*, and our study represents a snapshot of this evolutionary back-and-forth. Clearly, more research on this complex system is needed to better understand the evolutionary ramifications of this multi-species interaction, including behavioral and physiological adaptations of the hosts (especially *N. triops*) and the parasitoid, host quality and resulting fitness effects on the fly. The current findings provide important starting points to guide future research on the parasitoid and its hosts.

## Figures and Tables

**Figure 1 insects-14-00744-f001:**
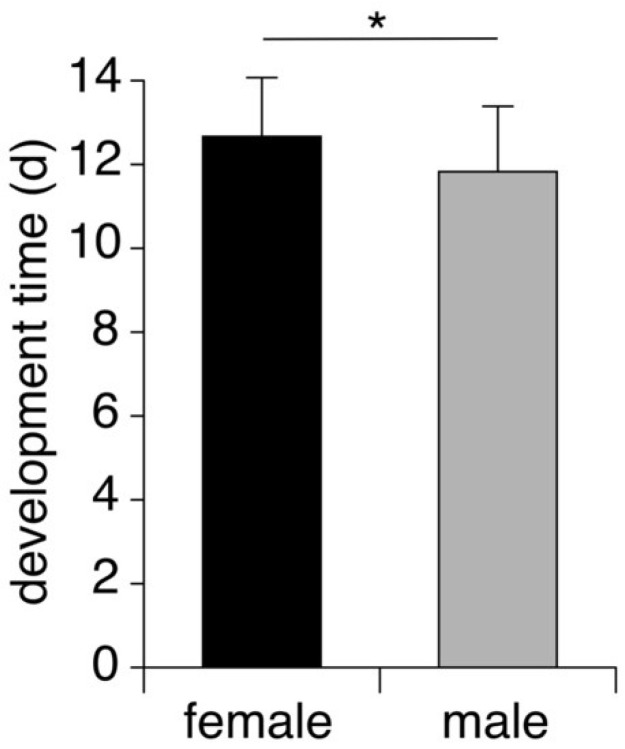
Average time (±S.D.) *O. lineifrons* pupae took to develop into male (n = 29) and female (n = 24) adult flies across all four host species. Asterisk indicates a significant difference.

**Figure 2 insects-14-00744-f002:**
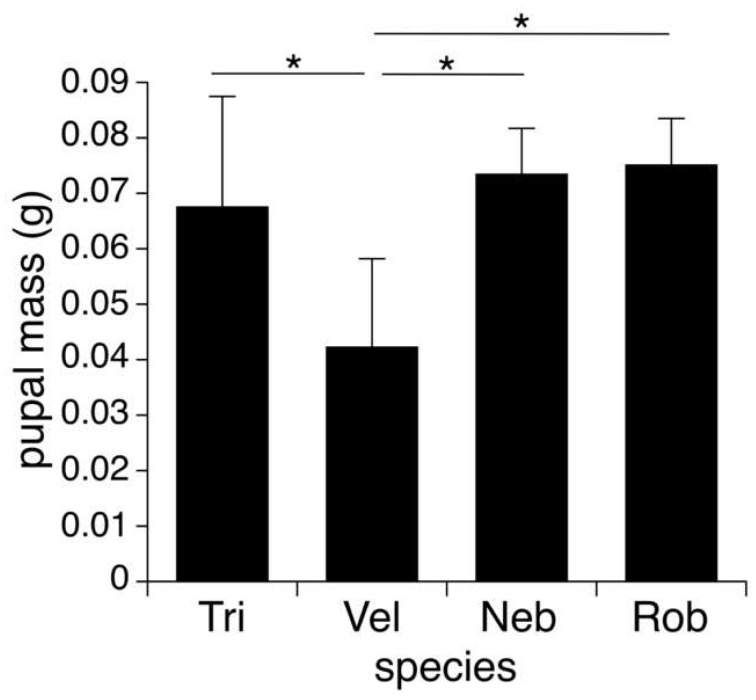
Average mass (±S.D.) of pupae emerging from *N. triops* (Tri; n = 19), *N. velox* (Vel; n = 13), *N. nebrascensis* (Neb; n = 4), and *N. robustus* (Rob; n = 8) hosts and successfully developing into adult flies. Asterisks indicate significant differences.

**Figure 3 insects-14-00744-f003:**
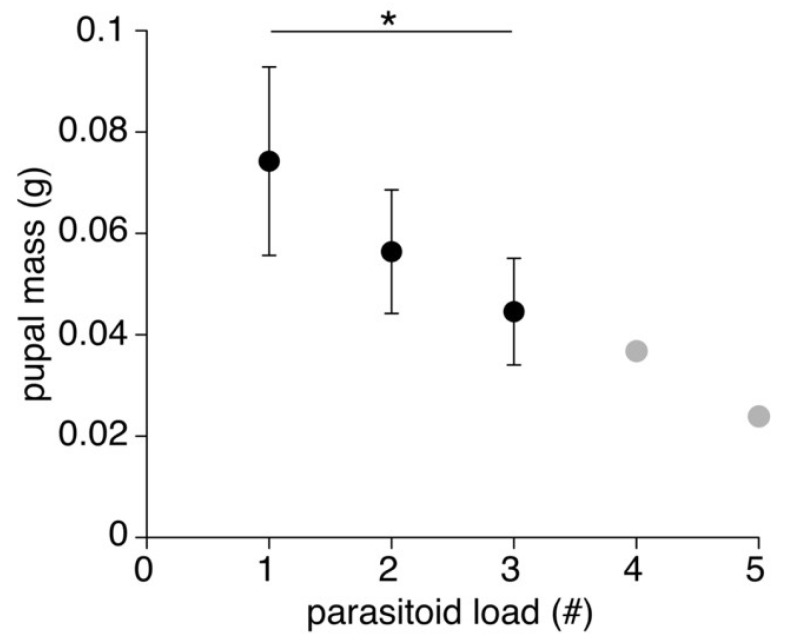
Average mass (±S.D.) of pupae that successfully developed into adult flies across hosts with different parasitoid loads (n = 8–23). Grey symbols indicate masses with low sample sizes (4–5 parasitoids/host, n = 1–2) and are only shown for completion. # indicates the number of parasitoids per animal. Asterisk indicates significant difference.

**Figure 4 insects-14-00744-f004:**
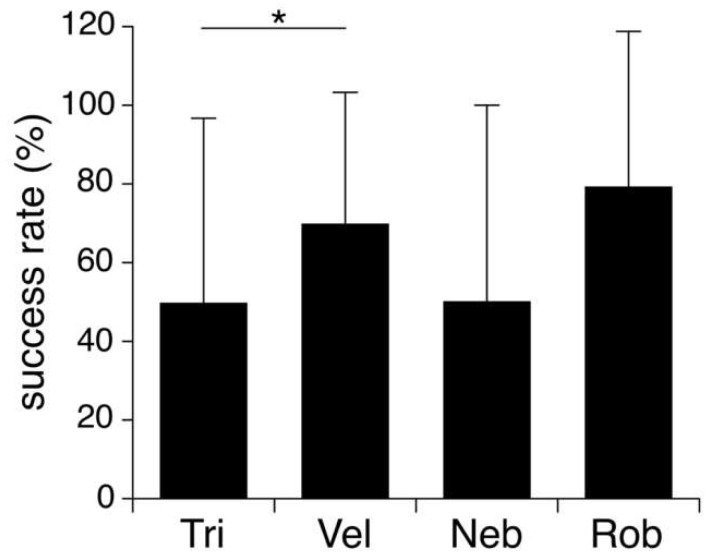
Average developmental success rate (±S.D.) of pupae emerging from *N. triops* (Tri; n = 44), *N. velox* (Vel, n = 23), *N. nebrascensis* (Neb, n = 6), and *N. robustus* (Rob, n = 12) hosts developing into adult flies. Asterisk indicates significant difference.

**Figure 5 insects-14-00744-f005:**
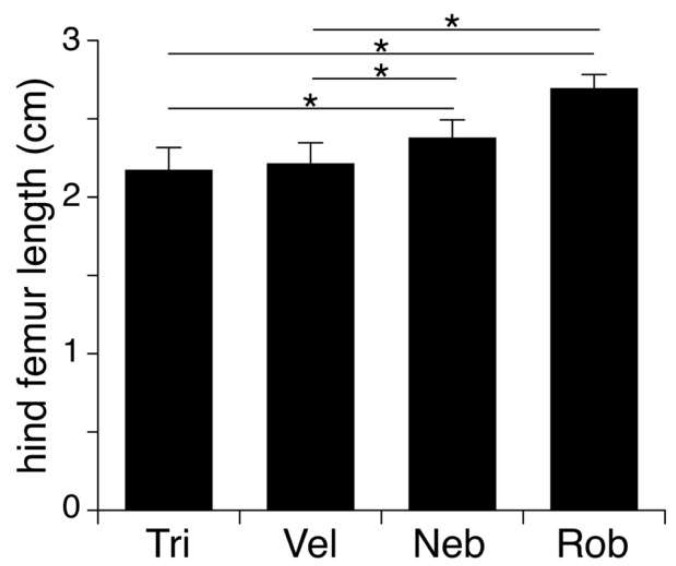
Average hind femur length (±S.D.) of *N. triops* (Tri, n = 46), *N. velox* (Vel, n = 5), *N. nebrascensis* (Neb, n = 26), and *N. robustus* (Rob, n = 25) hosts. Asterisks indicate significant differences.

**Table 1 insects-14-00744-t001:** Mean development time (±S.D.) and clutch size of *O. lineifrons* larvae in different host species and cumulative parasitism rate of each host species. Sample sizes are indicated in parentheses.

Species	Development Time (mean ± S.D.)	Clutch Size (mean ± S.D.)	Cumulative Parasitism Rate (%)
*N. triops*	12.33 ± 1.1 (25)	1.93 ± 0.97 (44)	31.8% (154)
*N. velox*	12.51 ± 3.0 (21)	2.30 ± 1.18 (23)	73.0% (33)
*N. nebrascensis*	12.50 ± 1.9 (4)	1.17 ± 0.41 (6)	17.5% (40)
*N. robustus*	12.10 ± 1.0 (10)	1.42 ± 0.67 (12)	14.3% (84)

## Data Availability

All data are available upon request.

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
