# Peer review of "Multi-Species Host Use by the Parasitoid Fly *Ormia lineifrons"

_insects, 2023, doi:10.3390/insects14090744_

Round 1
Reviewer 1 Report
Rogers & Beckers report the results of a study on a tachinid fly that attacks four species of katydids in Kentucky, USA. The manuscript has an obvious evolutionary focus. I believe the study would be of interest to readers of Insects, but requires improvement before it is suitable for publication.
Main points.
The authors use the term "success" when reporting parasitism throughout the manuscript. In some parts it is clear what they mean by success, in other parts less so. I would ask the authors to revise their use of this term. If they are describing the ability of the parasitoid to complete its development in a particular host species, perhaps use "ability to develop" or host suitability or a similar term. The authors mention in the Discussion that they have no information on larval survival in hosts, so "success" may not be the most suitable term.
From the beginning I was looking for information on the prevalence of parasitism of the different species. The authors should begin the Results section with this info.
I have made suggestions for improvements to the text and typos on a scanned copy of the manuscript.
Numbered points (see scanned copy of manuscript)
1. I think that you need to make it clear from the outset that the different host species are available sequentially over time, otherwise one starts by thinking that this system involves a frequency dependent selection issue.
2. Text needs rewording.
3. Again, please make it clear that hosts are available sequentially over time.
4. Lines 29 – 32 need rewording; N. tripos did not display lower success – the parasitoid was less 'successful (?)' when developing in N. tripos. The same for the other species.
5. But this effect is likely to be highly constrained by the need to parasitize the other species, presumably?
6a. I would like to see information (perhaps in Table 1?) on the mean prevalence of parasitism of each of the hosts attacked by this fly. It seems from the sample sizes that parasitism can vary (lower in Nn and Nr and higher in Nt and Nv?)
6b. Saturated methylparaben solution? Could you provide an estimate of the concentration? This compound is typically dissolved in ethanol before being made into aqueous solution.
7. Were the data checked for normality and equality of variance prior to ANOVA?
8. This information on prevalence of parasitism needs to be moved to the beginning of the Results section (and could be usefully presented in Table 1, or in a separate figure).
9. This section presents information on host size, but fails to test whether host size affected developmental "success". How should success be measured here....clutch size?...offspring weight? Also, I note that it would also be interesting to compare host size effects within each species (among individuals of different sizes) and between species, would it not?
10. This text follows from the previous text. Please delete.
11. The argument regarding N. triops as a poorer quality host compared to the other species needs to be framed in terms of the prevalence of parasitism. The high parasitism of Nv contrasts with the low parasitism of Nn and Nr – presumably this has a marked effect on selection in the host, doesn't it? Also, is Ormia an important natural enemy compared to other predators/parasites/pathogens? If 90% mortality is due to bird predation or fungal infection then selection for resistance/avoidance of Ormia would be negligible, wouldn't it?
12. References should be formatted for Insects.

Some typos and grammatical errors.
Author Response
We would like to thank the reviewer for the productive and insightful comments. Below, we address each comment of the reviewer (responses in italics) and indicate in the revised manuscript (track/changes are visible in attached document, attached) where the changes can be found.
Rogers & Beckers report the results of a study on a tachinid fly that attacks four species of katydids in Kentucky, USA. The manuscript has an obvious evolutionary focus. I believe the study would be of interest to readers of Insects, but requires improvement before it is suitable for publication.
Main points.
The authors use the term "success" when reporting parasitism throughout the manuscript. In some parts it is clear what they mean by success, in other parts less so. I would ask the authors to revise their use of this term. If they are describing the ability of the parasitoid to complete its development in a particular host species, perhaps use "ability to develop" or host suitability or a similar term. The authors mention in the Discussion that they have no information on larval survival in hosts, so "success" may not be the most suitable term.
We want to thank the reviewer for this comment. We clarified what we refer to in terms of ‘success’ throughout the manuscript. It is the percentage of pupae developing into adult flies. We hope that specifically indicating what we mean throughout the manuscript clarifies this point. In addition, we more explicitly addressed the issue of larval development in the discussion for transparency to the reader (lines 438-442).
From the beginning I was looking for information on the prevalence of parasitism of the different species. The authors should begin the Results section with this info.
We moved the parasitism data at the beginning of the results section as suggested (lines 300-306).
I have made suggestions for improvements to the text and typos on a scanned copy of the manuscript.
Numbered points (see scanned copy of manuscript)
Thank you for the improvements and we changed them as suggested throughout the manuscript.
I think that you need to make it clear from the outset that the different host species are available sequentially over time, otherwise one starts by thinking that this system involves a frequency dependent selection issue.
We changed the wording to make this point clearer (lines 27-28).
Text needs rewording.
We changed the wording to make this point clearer (lines 34-36).
Again, please make it clear that hosts are available sequentially over time.
We changed the wording to make this point clearer (line 70).
Lines 29 – 32 need rewording; N. tripos did not display lower success – the parasitoid was less 'successful (?)' when developing in N. tripos. The same for the other species.
The reviewer is correct. Thank you. We reworded the statements accordingly (lines 73-76).
But this effect is likely to be highly constrained by the need to parasitize the other species, presumably?
We don’t know what causes the poor performance of fly pupae emerging from N. triops (see discussion) and its very likely that all hosts share very similar physiologies, i.e., there does not need to be tradeoff with other host species. For example, if the amount of fat tissue differs in N. triops compared to N. velox (or other species) because of their adult diapause, as we discuss, this can cause differences in host quality. This in turn would select for better utilization of the limited host resources without constraints imposed by other host species, since being more effective in using resources would be beneficial for the fly across hosts. Or in other words, adaptations to one host may not be held in check by opposing adaptations related to other hosts. We reworded our statement slightly (lines 80-81).
6a. I would like to see information (perhaps in Table 1?) on the mean prevalence of parasitism of each of the hosts attacked by this fly. It seems from the sample sizes that parasitism can vary (lower in Nn and Nr and higher in Nt and Nv?)
We added the parasitism rates across the fly generations from the previous study in the Methods to better explain the parasitoid/host interactions (lines 222-224). We added the percentages of the cumulative parasitism rates in table 1.
6b. Saturated methylparaben solution? Could you provide an estimate of the concentration? This compound is typically dissolved in ethanol before being made into aqueous solution.
We added this information to the text (line 232). We did not use ethanol to dissolve the Methylparaben but stirred it into hot water.
Were the data checked for normality and equality of variance prior to ANOVA?
Yes, they were. One data set (developmental time difference between male and female flies) was not normally distributed, and transformations did not help either. We ran a bootstrap method (residual bootstrap by Efron with 1000 rounds of generating sample data based on our data residuals) to the p-value, which still indicated a significant difference. We added this information to the methods section (lines 290-296).
This information on prevalence of parasitism needs to be moved to the beginning of the Results section (and could be usefully presented in Table 1, or in a separate figure).
Changed as suggested.
This section presents information on host size, but fails to test whether host size affected developmental "success". How should success be measured here....clutch size?...offspring weight? Also, I note that it would also be interesting to compare host size effects within each species (among individuals of different sizes) and between species, would it not?
We agree with the reviewer and phrased this analysis and the results more carefully (lines 256). Based on the comment, we were not sure which comparisons the reviewer would like to see (weight affecting which parameter of the fly?). However, we considered a similar analysis before, i.e., the effect of host size on pupal development success, but ran into several issues. First, data on host size was only measured in one year (2020), and for 2 of the 4 hosts we only have data (both success rate and host size) for 4 animals each. Second, individual hosts had a range of parasite loads and as our figure 3 shows, pupal size decreases with parasite load, suggesting increased competition with increasing parasite load. So, interpreting the effect of host size on e.g., pupal mass or developmental success of the fly is difficult because parasite load likely affects pupal mass and/or developmental success as well. Third, even though we have pupal mass data, not all pupae developed into adult flies further complicating the analysis, i.e., a lack of development of some pupae might not be because of the host size, but competition before emerging from the host, or other reasons. Nevertheless, we did run a mixed linear model with host size, parasite load, host species, and their interactions as main effects and pupal development success rate as response variable, finding no significant effect of host size or species or the interaction of the two, which could be because of the reasons outlined above. We did not report this analysis in the revision because of the possible confounding effect of parasite load and small sample sizes for two species. However, if the reviewer would like us to show this analysis related to host size, we are happy to do so in the revision.
This text follows from the previous text. Please delete.
Changed as suggested (line 402).
The argument regarding N. triops as a poorer quality host compared to the other species needs to be framed in terms of the prevalence of parasitism. The high parasitism of Nv contrasts with the low parasitism of Nn and Nr – presumably this has a marked effect on selection in the host, doesn't it? Also, is Ormia an important natural enemy compared to other predators/parasites/pathogens? If 90% mortality is due to bird predation or fungal infection then selection for resistance/avoidance of Ormia would be negligible, wouldn't it?
Neoconcephalus are nocturnal and hide (very well) from predators during daytime. Besides bat predation when in flight (no data on the predation rate though), parasitism by the acoustically orienting O. lineifrons is the major predation risk that Neoconcephalus encounters. Weekly parasitism rates for N. velox and N. triops reach 100% over the course of the fly/katydid seasons and 40 and 60% for the other two species (we added this information to the text: 222-224). We were not quite sure about how to frame host quality in terms of prevalence of parasitism, as suggested by the reviewer. Even though the cumulative prevalence differed across species, it cannot be used to infer host quality, since the parasitoid has to use each sequentially occurring host. Or in other words, if multiple hosts were available at the same time, I could see that parasitism prevalence could reflect host quality (i.e., better host species have higher prevalence), but this is not the case in our system. We added some discussion related to the different selective pressures on the hosts in the revised manuscript (lines 486-506).
If we misunderstood this comment, we kindly ask for more guidance so that we can better address this point in another revision.
References should be formatted for Insects.
We formatted the references in the revised version.

Reviewer 2 Report
My comments are in the attached Word doc. The science is good and the English is also fine.

Author Response
We would like to thank the reviewer for the productive and insightful comments. Below, we address each comment of the reviewer (responses in italics) and indicate in the revised manuscript (track/changes are visible in attached document) where the changes can be found.
Introduction
Line 52 and following—Here is seems necessary to me for the authors to distinguish two cases, which in the text are conflated. Case I is, as in this study system, a parasitoid has a series of host that are used sequentially over time, such that any single poor host restricts populations of the parasitoid in the next host. Case II is that of parasitoids with many hosts, but which are not sequential over time. In Case I it is reasonable to assume that a temporally-required, but poor host might be exerting important selection pressure on the parasitoid. Conversely in Case II, the poor hosts likely do not exert such pressure because parasitoids usually have better hosts to choose from. Me
Thank you for this comment. We added this information in the introduction as suggested (lines 163-168)
Methods
Nothing
Results
Fig. 2 on line 190 seems to be the heart of the paper, i.e., that one species produced smaller parasitoids.This then suggests, but does not prove, that this matters in the overall ecology of the parasitoid across its generations. The obvious next step would be to bring this poor species and the best host species into laboratory culture and rear the fly in various sequences (such as “A” a string of best, “B” a string of worst, and “C” alternating best and worst). If the poorest host is restraining the fly then the alternating scenario C should reduce fitness in comparison to the constant best scenario (A).
Even though Figure 2 is important, we believe that Figure 3, i.e., success rate of pupal development to adult flies is the more important evidence that the host quality differs across the hosts and thus likely affects the ecology and evolution of the parasitoid. Note that in Figure 2, N. velox has an even lower pupal mass than N. triops, but has a higher success rate of pupae developing to adults compared to N. triops (Figure 3.) We discuss this seemingly paradoxical result in detail [lines 376ff]. The proposed experiment would indeed be insightful, and we thank the referee for this great idea for a future experiment.
Discussion
Line 230 a grammatical error: before “however” use a semicolon, not a comma, and then add a comma after the “however.”
Changed as suggested (line 364).
Line 247 if N. Florida is a species name, it should be N. florida. If that is not what is meant here, there is something left out, or perhaps N. was meant to be the word “in”
Changed as suggested (line 390), i.e., ‘N’ was not supposed to be there.
For this story to be convincing, information (in future studies) is needed on the mechanism that make the poorest host poor.
The lower success rate for developing pupae (Figure 3) is a strong indicator that parasitoids that develop and emerge from N. triops hosts are presumably in poor condition and the host is negatively affecting the pupal development. We added a statement in the text that future studies are required to better understand host quality differences (lines 584-587).
Finally an article by Martine Aluja in INECOL in Jalapa, Mexico comes to mind in which the key fruit fly pest of mango has a string of generations in alternative fruit hosts in which its survival varies based on variable survival in big vs small fruits due to higher parasitism by a guild of generalist parasitoids, which can reach a larger percentage of hosts in a fruit in small species of fruits, while in big fruits some larvae are out of reach of the parasitoids ovipositor.
Aluja, M., J. Sivinski, R. Van Driesche, A. Anzures-Dadda, and L. Guilen. 2014. Pest management through tropical tree conservation. Biodiversity and Conservation 24 (3): 831-853.
Thank you for suggesting this manuscript. We added this citation [20 in the text] in the revised version of the manuscript (lines 149-160 and 166, 485).

Round 2
Reviewer 1 Report
Some errors remain.
L2. Italicize species name.
L 93. Genus name is Anastrepha (not Anestrepha)
L 153. ~0.4% water/Methylparaben solution - - change to 0.4% methylparaben solution.
L205. F value is missing the treatment and error degrees of freedom.
L424. Delete " The cumulative parasitism of N. "
L244. Delete "3." at start of sentence.
L244-45. There is no Figure 6. Change to Fig 5?
L254. I found both rats and rage in this sentence (!).
L338. Delete "Our study suggests that N."
L355. Delete comma after "future"
L450. Delete "28".
L454. Delete "30."
L478. Delete "43"
Some typos.
Author Response
I want to thank the reviewer for catching the mistakes/typos; I did not catch some of them. Please note that some of the correctly identified issues must have been accidental mistakes when the manuscript that I uploaded with the track/changes highlighted to the PDF that was sent out for review, i.e., there were some items in the PDF that were supposed to be deleted. To prevent this from happening again, I highlighted the changes in yellow in the manuscript for the reviewer to find. The line numbers refer to the uploaded Word file. The responses to each item are in italics below each comment. Again, thank you.
L2. Italicize species name.
Changed as suggested.
L 93. Genus name is Anastrepha (not Anestrepha)
Changed as suggested (line 116).
L 153. ~0.4% water/Methylparaben solution - - change to 0.4% methylparaben solution.
Changed as suggested (line 178).
L205. F value is missing the treatment and error degrees of freedom.
I talked to my statistician about this when we did this analysis. He replied that there are no degrees of freedom, as the p-value is determined as the proportion of bootstrapped test statistics that exceed the actual test statistic, not by using the F distribution. So, there are no DF when bootstrapping the p-value (Methods 212-215).
L424. Delete " The cumulative parasitism of N. "
Here we are not sure which sentence the reviewer refers to because of the line number. We found one sentence that started with these words but we were not sure what to delete, i.e., if we delete this part of the sentence (or the whole sentence?), then that sentence/section would not read well. We kindly ask for more advice on how we can address this issue.
L244. Delete "3." at start of sentence.
This number was deleted in the revision but may have accidentally been shown in the PDF that was sent out as a result of converting the track/changes showing Word document to the PDF document. It was not supposed to be there.
L244-45. There is no Figure 6. Change to Fig 5?
The reviewer is correct. Thank you. (line 249)
L254. I found both rats and rage in this sentence (!).
My apologies (line 257). Neither rats nor rage should have been in this sentence.
L338. Delete "Our study suggests that N."
This sentence was deleted in the revision but may have accidentally been shown as a result of converting the Word to the PDF document. It was not supposed to be there.
L355. Delete comma after "future"
Changed as suggested (line 357).
L450. Delete "28".
This number was deleted in the revision but may have accidentally been shown in the PDF that was sent out as a result of converting the track/changes showing Word document to the PDF document. It was not supposed to be there. .
L454. Delete "30."
This number was deleted in the revision but may have accidentally been shown in the PDF that was sent out as a result of converting the track/changes showing Word document to the PDF document. It was not supposed to be there.
L478. Delete "43"
This number was deleted in the revision but may have accidentally been shown in the PDF that was sent out as a result of converting the track/changes showing Word document to the PDF document. It was not supposed to be there.